# Food Habits and Lifestyle of Romanians in the Context of the COVID-19 Pandemic

**DOI:** 10.3390/nu14030504

**Published:** 2022-01-24

**Authors:** Valentin Năstăsescu, Magdalena Mititelu, Tiberius Iustinian Stanciu, Doina Drăgănescu, Nicoleta Diana Grigore, Denisa Ioana Udeanu, Gabriela Stanciu, Sorinel Marius Neacșu, Cristina Elena Dinu-Pîrvu, Eliza Oprea, Manuela Ghica

**Affiliations:** 1Department of Physical and Colloidal Chemistry, Faculty of Pharmacy, University of Medicine and Pharmacy Carol Davila, 020956 Bucharest, Romania; valentin.nastasescu@umfcd.ro (V.N.); cristina.dinu@umfcd.ro (C.E.D.-P.); 2Department of Clinical Laboratory and Food Safety, Faculty of Pharmacy, University of Medicine and Pharmacy Carol Davila, 020956 Bucharest, Romania; diana-nicoleta.grigore@rez.umfcd.ro (N.D.G.); denisa.udeanu@umfcd.ro (D.I.U.); 3Press Office, Ovidius University of Constanța, 900527 Constanța, Romania; tiberius.stanciu@365.univ-ovidius.ro; 4Pharmaceutical Physics and Informatics Department, Faculty of Pharmacy, Carol Davila University of Medicine and Pharmacy, 020956 Bucharest, Romania; doina.draganescu@umfcd.ro; 5Department of Chemistry and Chemical Engineering, Faculty of Applied Sciences and Engineering, Ovidius University of Constanța, 900527 Constanța, Romania; 6Proffesional Farma Line, 116 Republicii Street, 105200 Baicoi, Romania; sorinel.neacsu@pfarma.ro; 7Department of Organic Chemistry, Biochemistry and Catalysis, Faculty of Chemistry, University of Bucharest, 030018 Bucharest, Romania; eliza.oprea@g.unibuc.ro; 8Department of Biostatistics, Faculty of Pharmacy, University of Medicine and Pharmacy Carol Davila, 020956 Bucharest, Romania; manuela.ghica@umfcd.ro

**Keywords:** COVID-19 pandemic, SARS-CoV-2, diet changes, lifestyle changes, social life changes, psycho-affective changes

## Abstract

The pandemic caused by the SARS-CoV-2 virus has produced significant changes in socio-cultural life, diet, and interpersonal relationships across the world’s population. The present study aims to identify changes in lifestyle and diet among the Romanian population one year after the onset of the COVID-19 pandemic. An online questionnaire with 58 items (addressing the following aspects: socio-demographic and anthropometric data, current eating habits, and lifestyle changes) was distributed using institutional mailing lists and social media. A total number of 2040 respondents participated in the study, of whom 1464 were women, and 576 men. Among the respondents, 1598 came from urban areas and 442 from rural areas. The processing of the collected data showed significant changes in the behavior of the respondents caused by the pandemic situation with psycho-affective changes in some cases. The number of people who had anxiety, depression and nervousness increased by up to 20%. The majority of respondents (over 57%) were up to 30 years old, either students (43.50%) or employees going to workplaces (33.20%). Analyzing eating habits, we found that diet modification was needed to increase the daily consumption of fruits, vegetables, fish, and seafood. Regarding weight status during the pandemic, we noticed that 34.7% of normal-weight respondents said that they gained weight while 49.7% of overweight people and 52.5% of obese people said that they gained weight (*p* < 0.0001). Regarding psycho-emotional behavior, 11.81% of the surveyed women stated that they frequently had depressive states during the pandemic period and 11.63% of the men stated that they frequently presented depressive states during the pandemic (*p* = 0.005).

## 1. Introduction

The crisis generated by the SARS-CoV-2 virus requires solutions with joint efforts from all states to protect population health and reduce the damage to national economies. In order to avoid catastrophic outcomes for national health systems and reduce the number of casualties, many countries have initiated social distancing policies to slow the spread of the SARS-CoV-2 virus. The COVID-19 pandemic has caused all employees worldwide to suddenly experience significant changes, both in the family and in their professional activity, the most affected being low-wage workers without access to social protection [1,2,3].

The economic impact of the coronavirus crisis varies from one industrial sector to another and depends on a number of factors, including the possibility of adapting to supply chain disruptions, the existence of stocks or dependence on the capacity of production processes. The tourist ecosystem has also been severely affected by traffic and travel restrictions due to the coronavirus crisis [4,5,6].

For many children and parents, the COVID-19 pandemic has changed daily routines. As more and more families are facing social distancing, self-quarantine and the closure of schools and jobs, they may feel stressed. Stress occurs not only because of health concerns, but also because families have to deal with a new reality, when they have suddenly found themselves at home for several days with children whose daily routines were severely affected. Isolation at home and limited interactions with their peers make children anxious. The lack of outdoor activities could make them more agitated and frustrated, stressing their parents even more [7,8].

In all countries of the world, in the conditions of isolation at home, limited contact with members of the social network, working from home, closed schools/universities, the lack of facilities for spending time and financial insecurity, family relationships have become increasingly strained, sometime reaching extreme manifestations [9,10,11,12,13].

The lockdown, travel restrictions, and economic problems have made people feel lonelier in 2020 than ever before. Moreover, a lack of exercise and increased compulsive eating during the pandemic has led to increased obesity among the population, with serious health implications [14,15,16,17].

Romania was facing a health crisis generated by the fourth wave of the COVID-19 pandemic. As of 4 October 2021, 1,274,119 confirmed cases of SARS-CoV-2 virus infection were reported in Romania, of which 1,130,791 (88.75%) were cured patients and 37,677 (2.96%) were deceased patients. Regarding the Vaccination Campaign, as of the same date 10,428,307 doses of SARS-CoV-2 virus vaccine were administered and 5,434,368 people were reported to be immunized, meaning 27.99% of the total population [18].

Despite all of the efforts made by the authorities, the vaccination campaign of the Romanian population did not make much progress; in addition, the government is reluctant to adopt the normative act regulating the implementation of the green COVID-19 certificate at work.

Regarding the support of the population for better access to basic foods, Romania, following the example of Poland, is preparing to take steps within the European Union to approve the reduction of value-added tax to zero for basic products (meat, eggs, products dairy products).

In the current context caused by the COVID-19 pandemic, the population is becoming more and more aware of the need to adopt a healthy lifestyle and a balanced diet, primarily to support the immune system. Strong immunity is the best barrier against virus infections, including those with the new coronavirus. The body’s immunity is affected by a large number of factors. Stress diminishes the body’s immune response to pathogens. Food quality is also an important factor that affects the body’s reserves and endurance. The health of the body depends on the lifestyle that a person adopts. Factors that influence a healthy lifestyle are diet, hydration, sleep time and physical activity. The quality of sleep has an impact on both the state of health and the psyche. Therefore, sleep hours must be monitored, and a balanced sleep program is needed, without excesses and without radical changes. One way to strengthen the body’s endurance is physical exercise, performed consistently. Moreover, exercise also contributes to maintaining a good mental tone. A healthy lifestyle includes following a program that includes three main meals each day, as well as eating healthy foods, rich in vitamins. Additionally, any excess should be avoided, especially when it comes to unhealthy foods or drinks [19,20,21,22].

Various clinical studies suggest the need to consume 20–30 different types of food during a week, particularly of plant origin. Food diversity compensates for nutrient deficiencies and reduces the toxic effect that excessive intake of natural or processed compounds from food can have. The intake of proteins, carbohydrates, lipids, vitamins and minerals must cover the body’s energy needs and structural needs. The most important sources of essential amino acids (complete proteins) are milk, meat, eggs, and cheese. The portion of food must be modified to favor the variety of food, in order to cover the dietary needs of the individual [23,24,25,26,27,28].

Additionally, certain drugs affect the body’s immunity. Repeated administration of antibiotics, steroid anti-inflammatory drugs and chemotherapy inhibits the immune system [29,30,31]. Prolonged exposure to toxic substances (air pollution, toxic work environment, various solvents, heavy metals) is another factor that affects the body’s ability to detoxify, regenerate and revitalize [32,33,34,35].

The objective of the paper is to investigate the changes in lifestyle, eating habits and psycho-affective state among the Romanian population one year after the onset of the COVID-19 pandemic.

## 2. Materials and Methods

### 2.1. Study Design

Between 3 May and 6 June 2021, a study was conducted among the Romanian population based on a questionnaire disseminated online using the Google Forms web survey platform to identify changes in lifestyle and diet during the first year of the COVID-19 pandemic. The link for the online survey was shared through social media and the institutional mailing lists of students and professional organizations from Romania. Participants were also asked to share the link of a questionnaire with their colleagues and friends. The questionnaire contains 58 items that address the following aspects: socio-demographic data (age, sex, occupation, area of residence, occupation), anthropometric data (height, weight), eating habits, lifestyle and psycho-affective behavior before and during the pandemic. The final database was downloaded as a Microsoft Excel sheet.

The criterion for participation in the study was an age over 18 years. Participation in the study based on the questionnaire was entirely voluntary and anonymous. Respondents were informed from the beginning about the purpose of the study and asked for permission to use and publish the data. The anonymous nature of the web survey does not allow sensitive personal data to be traced in any way.

Individuals who agreed to participate were informed of the purpose of the study and were assured of the confidentiality of the results. Additionally, the completion of the questionnaire was in compliance with GDPR rules.

### 2.2. Questionnaire Validation

In order to finalize the questions and eliminate the ambiguities, the questionnaire was previously distributed to a group of 200 people aged 18 years and above, in the form of a test together with an additional form with 6 questions related to: clarity of questions in the questionnaire, understanding of the questions asked, the layout of the questionnaire (framing of the questions, the format and size of the letters), the time required to complete the questions in the questionnaire, the relevance of the questions in relation to the purpose of the questionnaire and suggestions for possible improvements to the questionnaire. By addressing it, the aim was to identify any ambiguity in the questions, an ensure the correct completion. The results of testing the pilot phase questionnaire were analyzed by a group of 10 experts for content validation and optimization. Content validity ratio (CVR), and content validity index (CVI) were calculated [36,37]. The necessity of items was assessed using a three-point scale and scores ranged between −1 (not necessary), 0 (useful but not essential), and +1 (essential). The relevance and clarity of each item were also calculated using a four-point Likert scale: (1) not relevant/clear, (2) slightly relevant/clear and needs revision, (3) relevant/clear and needs minor revision, and (4) very relevant/clear. Irrelevant items were removed based on expert opinions and modifications were made to the remaining items to make them more accurate and increase clarity.

Cronbach’s α coefficient was used to determine the internal consistency of the questionnaire. The value of Cronbach’s α for our questionnaire came out to be 0.86, which suggests a good internal consistency and also that the scale in this study is reliable [38].

Finally, the final form of the questionnaire, presented in Appendix A, was designed.

### 2.3. Statistical Analysis

All the categorical characteristics from our study were analyzed qualitatively and were expressed as a percentage (%) and numbers (*n*). Additionally, the numerical characteristics were transformed into qualitative variables (such as age and BMI variables) and expressed by percentages (%) and numbers (*n*).

The Pearson’s chi-square ($\chi^{2}$) test was used to analyze the relationship between the categorical variables and thus to find patterns of dietary changes. Additionally, we used two-way analysis of variance (robust ANOVA) and multinomial logistic regression analyses were performed to analyze the factors (gender, age group, BMI group) that influenced the odds of assignment to the healthy diet, moderate healthy diet and unhealthy diet [39,40]. In order to analyze the adherence of the respondents to a healthy diet, the answers from questions 7–10 and 12–18 of the questionnaire were quantified by giving a score from 1 to 5. A healthy diet consisted of a daily intake of fruits and vegetables, a low consumption of saturated fats and the avoidance of the consumption of hydrogenated fats, a consumption of foods rich in valuable nutrients (meat, eggs, fish, seafood, dairy products) according to the body’s energy needs, a low alcohol consumption, and moderate consumption of sweets and bread. The consumption of unhealthy foods or the lowest consumption of healthy foods rich in valuable nutrients was scored with 1, and 5 scored the consumption of the healthiest foods or the high frequency of the consumption of foods rich in valuable nutrients. Based on these answers, we formed a raw score that could then be scaled into a T-score (standardized) with a mean of 50.0 and standard deviation of 10.0 [41]. Finally, the results were divided into three categories: scores under 33.3 became an unhealthy diet, over 66.6 became a healthy diet and the rest of the values were framed as a moderately healthy diet. The results of multinomial logistic regression analyses are expressed as risk ratio (RR) and standard errors.

The dependent variable is “healthy eating”, and the reference group is “moderate healthy diet”.

Independent variables are:Gender where the reference group is “female”;BMI group where the reference group is “normal weight”;Age group where the reference group is young people in the range “[18,19,20,21,22,23,24,25,26,27,28,29,30]”.

For all statistical analyses, *p* < 0.05 was considered significant. Statistical analysis was implemented using the open-source software R (R version 4.1.1) [42].

## 3. Results

### 3.1. Socio-Demographic and Anthropometric Data

Following the dissemination of the questionnaire in the online environment, a total of 2054 responses were obtained, of which 2040 were valid answers (answers with incorrect anthropometric data were eliminated). Of the respondents, 71.61% were women (1464) and 28.39% men (576). According to the registered answers, 1598 (78.14%) of the persons who completed the questionnaire lived in urban areas and 442 (21.86%) in rural areas. The anthropometric data (weight and height) were used to calculate BMI by using the Quetelet equation (body mass (kg)/height (m^2^)) and interpreted according to the criteria of the World Health Organization [43]. Most of those who participated in the survey were normal weight (57.78%), 7.25% were underweight, 24.97% were overweight and 9.98% were obese [44]. The sociodemographic and lifestyle characteristics of participants are presented in Table 1.

One year after the onset of the pandemic, 39.8% of respondents said they had maintained a constant body weight, 38.4% said they had gained weight and 32.6% said they had an increased appetite. Of the participants in the questionnaire, 62.4% stated that they engaged in sports during the pandemic and 26.4% stated that they were quarantined due to infection with the novel coronavirus. We noted a negative change in the quality of life of the population caused by the COVID-19 pandemic; most respondents said that living standards deteriorated during the pandemic (63.9%) and only 12% said that life improved (Table 1). Of the people surveyed, 38.1% said they played sports at home during the pandemic, 22.5% outdoors and only 6% in the gym. Regarding the frequency of sports activities during the pandemic, the percentage of those who did not exercise at all, or that exercised slightly or very rarely, increased and the percentage of those who used to do sports daily before the pandemic decreased. Thus, 13.4% of respondents stated that they exercised daily during the pandemic and 15.9% stated that they used to exercise daily before the pandemic, while 26.8% stated that they exercised very rarely during the pandemic (compared with 25.6% before the pandemic) and 37.6% that they did not play sports at all during the pandemic (compared with 34.1% before the pandemic). There was a tendency towards a sedentary lifestyle which intensified in the pandemic.

### 3.2. Current Eating Habits

From the analysis of the data presented in Table 2, a higher tendency among men to consume foods with a lower nutritional value was noticed compared with women. The majority of respondents preferred to consume foods with moderate nutritional value and only a small percentage, less than 6%, frequently consumed foods with high nutritional value. According to the answers, the main type of used dietary fat was sunflower oil (63% of respondents said they use it the most when cooking); in general, most respondents consumed only one serving of vegetables and fruits a day, 65% of respondents stated that they consumed poultry meat the most, 42.5% consumed fish and seafood very rarely or not at all and 41% only once a week, 28.4% consumed sweets and pastries daily and only 15.8% very rarely or not at all, and 40% consumed dairy products daily. Related to food consumption, 27.2% of respondents said they ate excessively during the pandemic and 30% of respondents said that during the pandemic they consumed healthier food. Culinary preferences were dominated by home-cooked food (79.9%), and only 21.5% of respondents declared they consumed ordered food and served at home.

According to the results of logistic regression analyses, presented in Table 3, we noticed a higher adherence to a healthy diet for older age groups compared with younger respondents (older age groups had higher RR values for a healthy diet compared to younger age groups, but they were not statistically significant). At the same time, the logistic regression model confirms a lower adherence of male respondents to a healthy diet compared with female respondents.

After processing the data, we observed that the majority of underweight and normal-weight persons were female respondents, while obese people were mainly male respondents (Figure 1). Regarding the tendency to increase body weight during the pandemic, we noticed that 34.7% of normal weight respondents said that they gained weight, while 49.7% of overweight people and 52.5% of obese people said that they gained weight (*p* ˂ 0.0001).

Regarding alcohol consumption, the respondents were divided into five groups according to the frequency of consumption: in group 1 were included the people who consumed alcoholic beverages the most frequently (one or more daily servings), while in group 5 were persons who consumed alcoholic beverages very rarely or not at all (Figure 2). We found that females who consumed alcohol the most often were of normal weight, while males who consumed alcohol the most often were from obese people (*p* = 0.0002). A high alcohol content of alcoholic beverages is correlated with a high caloric intake. In the present study, we could presume that either normal weight people consume low-alcohol drinks daily and in small quantities, or they were very active people so the additional caloric intake of alcohol was not excessive.

Thus, the restrictive measures imposed by the authorities reduced the infection rate, but the psycho-emotional pressure during the pandemic had a negative impact on the physical shape of the population and increased the obesity rate and therefore, the number of vulnerable people (Figure 3a).

From the analysis of the answers related to the appetite changes during the pandemic period (Figure 3b), we found that 32.4% of the normal-weight people stated that they had an increase in appetite, 33.6% of the overweight people and 39.2% of the obese people (*p <* 0.0001).

### 3.3. Lifestyle and Psycho-Affective Changes

Regarding a request for psychologist advice during the pandemic, only 8.13% of female respondents (*p* ˂ 0.02) and 5.2% of male respondents responded positively (Figure 4a).

The pandemic smokers were divided into four groups: in group 1 were included those who reported smoking daily, in group 2 those who smoked 2–3 times a week, in group 3 those who smoked occasionally and in group 4 non-smokers (Figure 4b). The processed data show that 23.77% of female respondents stated that they smoked daily and 35.41% of male respondents (*p* < 0.0001). From the data processing we found that in general, in all four groups, the most affirmative responses related to the frequent presence of nervousness in the pandemic period were recorded among female respondents.

Depending on the duration of sleep during the pandemic period, the respondents were divided into four groups (Figure 4c): in group 1 were included people who said they had frequent insomnia (7.3% of females and 4.68% of males in the group), group 2 included people who said they slept less than seven hours a night, group 3 included people who said they slept between 7 and 9 h a night (61.47% of females and 56% of males in the group) and group 4 included people who said they slept more than 9 h a night (*p* = 0.0002). From the analysis of the recorded answers, we found that the people who declared that they presented frequent states of fatigue during the pandemic period were mainly from the female respondents and especially from the groups of those who rested either too much or too little during the night. Overall, 40.23% of female respondents stated that they frequently experienced fatigue during the pandemic and 24.82% of male respondents stated that they frequently had fatigue (*p* < 0.0001).

Regarding psycho-emotional behavior during the pandemic period (Figure 4d), 23.63% of the female respondents stated that they had frequent states of anxiety and among the male respondents 12.67% stated that they had frequent states of anxiety; 11.81% of the women surveyed stated that they frequently had depressive states during the pandemic period and 11.63% of the men stated that they frequently presented depressive states during the pandemic (*p* = 0.005). Less than half of the people who frequently presented anxiety and depression sought the advice of specialists. Additionally, from the answers received, we found that the percentage of people who often had anxiety, depression, and nervousness increased during the pandemic by up to 20%.

From the collected data it was noted that 47.47% of females said they did not yet intend to be vaccinated against COVID-19, while 26% said they had been vaccinated with one or two doses (*p* < 0.0001); 37% of males participating in the survey said they did not currently intend to be vaccinated and 25% said they had been vaccinated with one or two doses (Figure 5a).

Regarding the flu vaccination, 17.34% of female respondents (*p* = 0.758) and 18.05% of male respondents said they got the flu vaccine (Figure 5b).

Among the most consumed food products in the pandemic compared with the previous period, according to the answers in the questionnaire, were: fresh fruits and vegetables, sweets, coffee and tea, homemade bread and pastries, and dairy products (Figure 6).

On the other hand, among the least consumed food products in the pandemic compared with the previous period were fast food products and snacks, industrial bakery products and soft drinks (Figure 7).

During the pandemic period, 13.8% of the respondents stated that they added one or more main meals, 15.6% added one or more snacks between meals and 57.5% stated that they consumed between one and two liters of water per day.

The purchase of food and household products was mainly made from supermarkets (68.5%) and shops near the house. Although the majority of respondents came from young and active people, we found that only a small percentage (19.9%) said they bought products online (Figure 8).

Among the most purchased products during the pandemic period were: staple foods (sugar, flour, rice, oil, water, bread, pasta), vegetables and fruits, disinfectants and cleaning products, meat and sausages (Figure 9). Related to the decision to buy food 75.1% of the respondents stated that when purchasing a food product, they had taken into account mainly the quality and nutritional value.

Regarding the quality of life during the pandemic, 57.9% of the respondents stated that it was depreciated due to social distance. In fact, 74.7% respondents said that they lacked freedom of movement and 63.3% had difficulties in communication with family and friends (Figure 10). Among the behavioral changes produced by the COVID-19 pandemic, we particularly noted: monitoring health (30.3%) and adopting a more understanding attitude towards others (28.3%). However, there were also people who stated that they were isolated from others (18.3%).

The way of spending the holidays also underwent major changes; 49.3% of the respondents stated that they did not go on vacation and 38.9% said that they traveled much less than in the period before the pandemic.

The majority of respondents (61.7%) said they were clinically healthy (Figure 11). Of the 2040 who answered the questions, 315 (15.4%) stated that they did not know if they suffered from any disease, which means that they were not used to regularly monitoring their health. Only 5% of those surveyed stated that they were obese but after the processing of anthropometric data we found that the percentage of obese participants was double (Figure 11, Table 1), which means that half of them had not realized that they were not normal weight. During the current pandemic context, 64.8% of respondents said that they feared more for the health of those close to them and only 25.9% said that they feared for their health.

During the pandemic 75.4% of respondents said they spent most of their free time watching movies and TV shows (Figure 12). Among the favorite ways to spend free time during the pandemic were meetings with family or friends, reading and outdoor activities.

## 4. Discussion

The COVID-19 pandemic brought significant changes in people’s lifestyle, diet, and in psycho-emotional behavior. However, on the other hand, the fear for one’s own health and those close to oneself determined the closer monitoring of one’s health, paying more attention to the quality and nutritional value of food, and also to personal hygiene.

The interest in purchasing fresh fruits and vegetables during the COVID-19 pandemic, as an improvement in nutritional habits, was also highlighted by similar studies carried out in Romania, Italy and Spain. Up to 40% of the respondents stated they had a significant increase in their intake of heathy foods, with positive effects on tone and health [45,46]. Additionally, the studies revealed that the adult population is yet far from attaining, and moreover preserving, healthy eating behaviors. Permanent eating habits are difficult to maintain and there is still a great share of the population that did not report positive consequences in eating behavior.

The viral infection and its consequences are just part of the negative impact of the COVID-19 pandemic. Changes in eating habits related to lockdown, leading to increased intake of sugars, fats, and salt, and also overeating in response to negative feelings, were observed by studies carried out in different countries [47,48].

If in countries with a Mediterranean diet such as Italy and Spain it was found that during the pandemic the population paid special attention to the nutritional quality of food [49,50], realizing the need to improve the diet to strengthen the immune system, the Romanian population needs to improve both nutrition and lifestyle. The present study found a low consumption of vegetables and fruits, well below the ratio recommended by nutritionists, namely a daily consumption of three servings of vegetables and two servings of fruits. The majority of respondents who were accustomed to eating vegetables and fruits daily stated that the ration was one serving per day. Additionally, regarding the consumption of fish or seafood, the data of the study indicated a very low consumption trend, with most of the respondents stating that they usually consumed these foods very rarely or not at all. Nutritionists recommend consuming 2–3 servings of fish or seafood per week [51,52].

One solution would be to implement educational programs in the field of nutrition in schools and universities, as well as to intensify campaigns to promote a healthy lifestyle in the media. However, we note that nutritional counseling is needed to increase the population’s focus on a healthy lifestyle to strengthen immune function and prevent metabolic disorders. In general, people are aware of the need to improve the quality of food, but the application in practice is deficient.

In terms of lifestyle improvement, there is a need to implement programs to combat sedentary lifestyles. The pandemic has generally restricted the possibility of playing sports in gyms [53], but the data collected in this study show a reduced tendency to practice sports both before and during the pandemic. This is all the more worrying as the vast majority of respondents were young and active people. According to the rules of healthy living, it is recommended to practice a minimum of thirty minutes of physical exercise daily. A sedentary lifestyle has now become one of the major causes of obesity and its implications [54].

The restriction of freedom of movement led to social distancing, nervousness, anxiety, depression, and compulsive eating for some people, and therefore to the depreciation of the quality of life. Many people do not realize that they need specialized care for their worsening psycho-emotional state and health.

There is a need for counseling and help for vulnerable people to cope with the new changes, in the current context.

The deterioration of the psycho-emotional state caused by the perceived pandemic in the present study is also highlighted by numerous other similar studies undertaken in various regions of the world [55,56].

From the data collected based on the questionnaire, we found that there was a deterioration of the psycho-emotional state among the respondents caused by the pandemic. Given that the majority of respondents came from the working population, we can draw attention to the need for specialist support, especially in companies with many employees and in universities, to improve mental health, as indicated by similar articles [57,58,59].

The pandemic period also brought significant changes in the way we spend our free time, the vast majority of respondents stating that they usually spend their free time watching TV, and in terms of how they spend their vacation, with almost half saying that they have not traveled.

A lower vaccination rate at national level as well as a reduced adherence to a healthy diet and a reduced tendency to periodically assess the state of health among the Romanian population are one of the important causes that have led to an increase in the mortality rate for infections caused by SARS-CoV-2 virus compared to other European countries [18]. Many patients have stayed at home without contacting a health care professional until the aggravation of the disease. It is necessary to implement educational programs in the medical field at the national level in order to raise awareness among the population of the importance of a healthy lifestyle, a balanced diet and regular monitoring of health to prevent metabolic disorders, strengthen the immune system and functioning of the body in optimal conditions.

The limitations of the study: Although the distribution of the questionnaire did not limit the participation only for persons under 18 years of age, in the sample surveyed the percentage of males and persons over 65 years of age was quite low. In the case of the elderly, the reduced participation is explained by the fact that the dissemination of the questionnaire was carried out among active people through institutional emails. Regarding the low percentage of male respondents, an explanation could be their lower availability to complete the questionnaire.

## 5. Conclusions

In the present study, which included 2040 respondents, we analyzed several aspects of daily habits among the population of Romania during one year of the pandemic, compared with the period before. The respondents were mainly from the active population, especially young students or employees in private companies. Following data processing, a number of important changes were found, some with a positive impact, and others with a negative impact.

Positive impacts: The increase in home cooking and healthy food consumption, especially vegetables and fruits. Increased attention to the health and well-being of those close to them.

Negative impacts: Increased incidence of overweight or obese, and the limitation of human interactions with psychological consequences. Some respondents declared sleep problems, depression, anxiety, fatigue, increased alcohol consumption, cigarette smoking or compulsive eating. The overloads of the health system have also generated a limitation of the population’s access to basic medical services.

## Figures and Tables

**Figure 1 nutrients-14-00504-f001:**
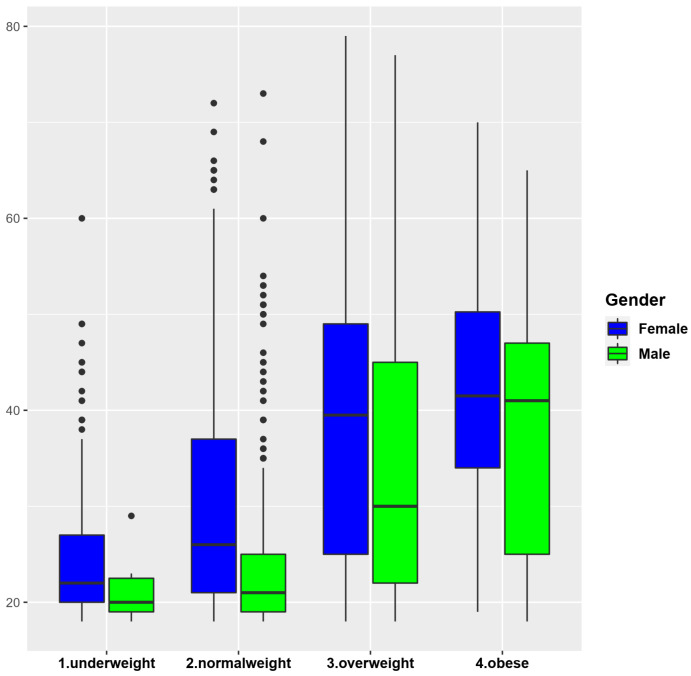
BMI according to gender and age.

**Figure 2 nutrients-14-00504-f002:**
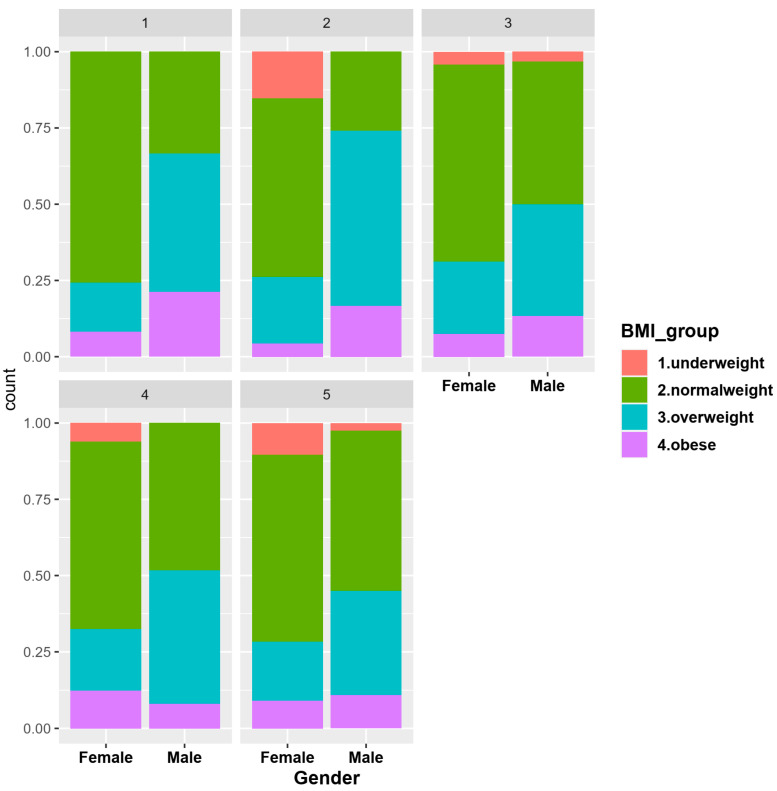
Alcohol consumption according to BMI and gender.

**Figure 3 nutrients-14-00504-f003:**
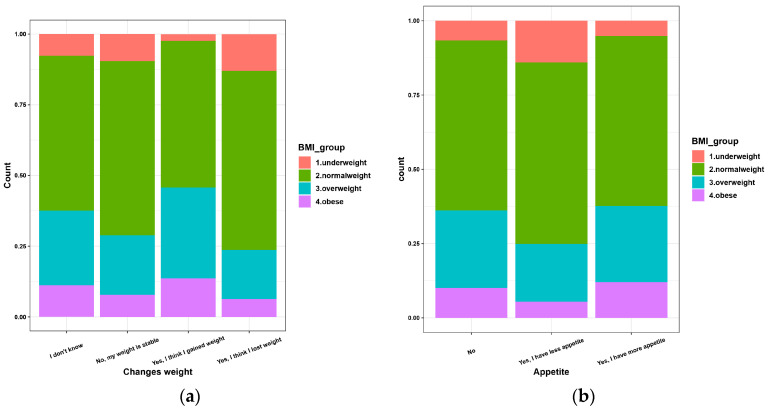
Perceptions regarding the changes in BMI during pandemic (**a**) and appetite changes (**b**) during pandemic according to BMI.

**Figure 4 nutrients-14-00504-f004:**
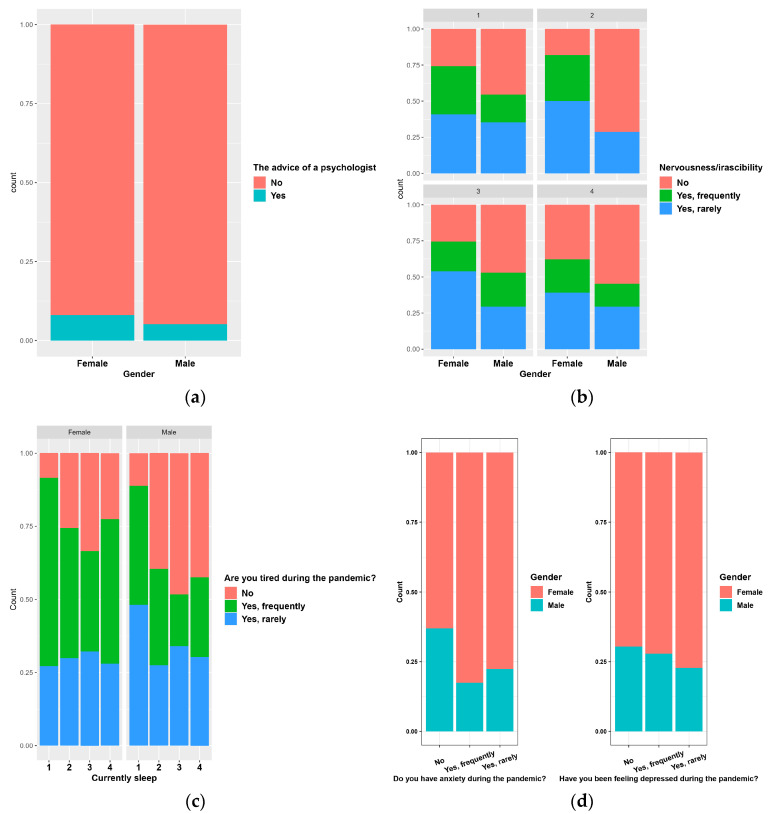
Lifestyle and psycho-affective changes. Persons requiring psychologist advice according to gender (**a**); smokers according to gender and nervousness/irascibility (**b**); tiredness related to sleeping habits according to gender (**c**); anxiety/depression according to gender (**d**).

**Figure 5 nutrients-14-00504-f005:**
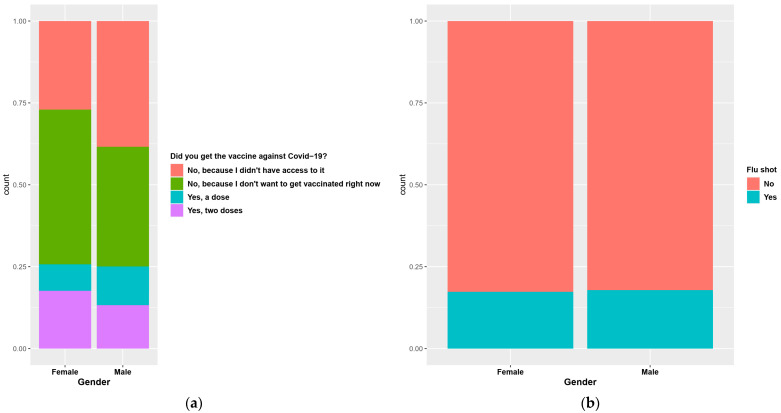
Vaccination among individuals according to gender: COVID-19 (**a**) and flu (**b**).

**Figure 6 nutrients-14-00504-f006:**
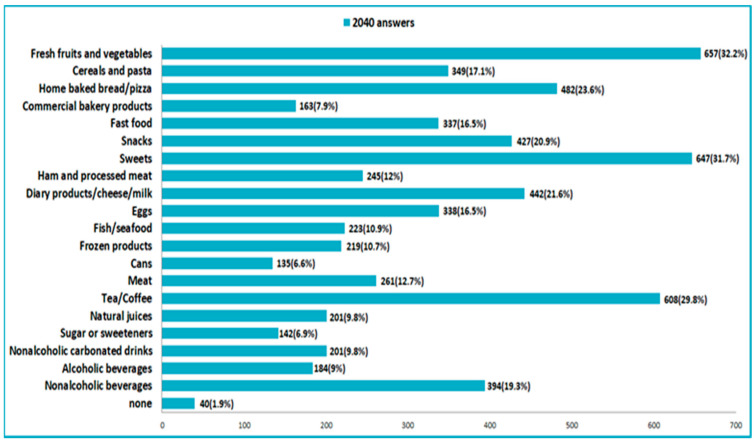
Food consumed more in the pandemic time.

**Figure 7 nutrients-14-00504-f007:**
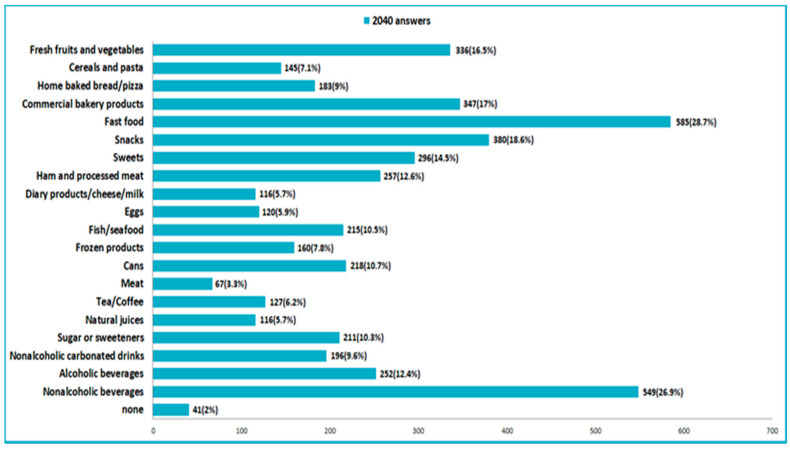
Food consumed less in the pandemic time.

**Figure 8 nutrients-14-00504-f008:**
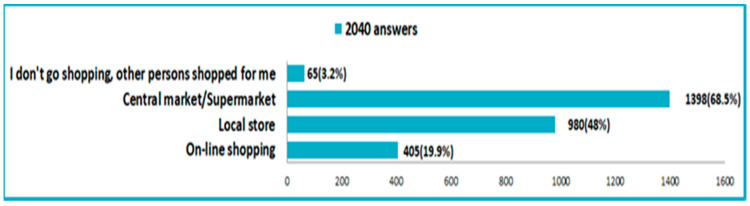
Ways of purchasing food and household products in the pandemic time.

**Figure 9 nutrients-14-00504-f009:**
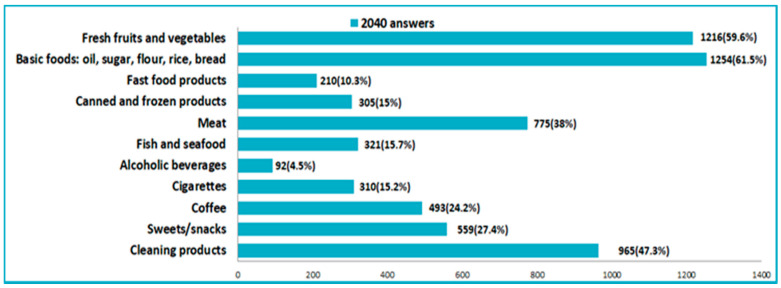
Products frequently purchased in the pandemic time.

**Figure 10 nutrients-14-00504-f010:**
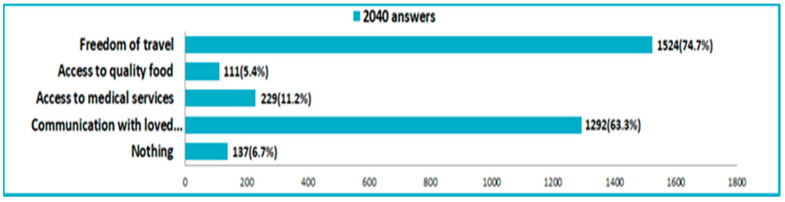
Deficiencies caused by the pandemic restrictions.

**Figure 11 nutrients-14-00504-f011:**
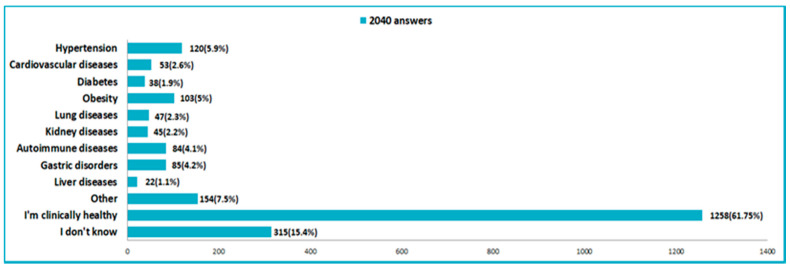
Distribution of chronic diseases of respondents in the pandemic time.

**Figure 12 nutrients-14-00504-f012:**
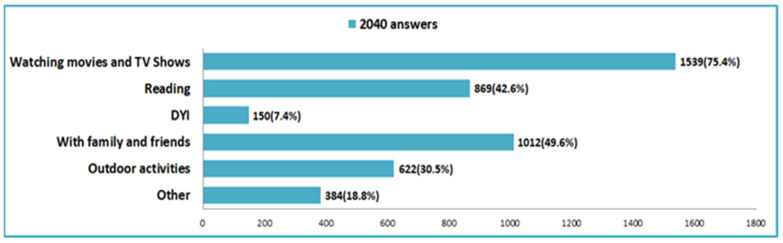
Ways to spend free time in the pandemic period.

**Table 1 nutrients-14-00504-t001:** Socio-demographic and lifestyle characteristics of participants (*n* = 2040).

Characteristics	*n*	%
*Gender*		
Male	576	28.3
Female	1464	71.6
*Age (years)*		
18–30	1166	57.5
31–50	684	33.5
51–65	155	7.6
>65	36	1.8
*Residence areas*		
Urban areas	1598	78.1
Rural areas	442	21.9
*Employment status*		
Unemployed	20	1.0
Socially assisted	0	0.0
Retired	65	3.2
Student	888	43.5
Housewife	43	2.1
I am going to work as usual	677	33.2
Working/studying from home	122	6.0
Hybrid work (teleworking and commuting)	180	8.8
I had my job temporarily suspended due to the pandemic	42	2.1
*Body mass index (BMI)*		
Normal limits (18.5–24.9)	1179	57.7
Overweight category (25–29.9)	509	24.9
Underweight category (<18.5)	148	7.2
Obese (≥30)	204	9.9
*Weight change during COVID-19*		
Weight is stable	811	39.8
Lost weight	328	16.1
Gained some weight	784	38.4
Do not know	117	5.7
*Appetite change during COVID-19*		
No	1065	52.2
Less appetite	309	15.1
More appetite	666	32.6
*Smokers during COVID-19*		
Yes	713	35.0
No	1327	65.0
*Sports practitioners during COVID-19*		
Yes	1272	62.4
No	768	37.6
*Alcohol consumer*		
*Very rarely or not at all*	1443	70.7
Daily	69	3.3
Moderate	528	26.00
*Quarantined due to COVID-19 infection*		
Yes	539	26.4
No	1501	73.6
*Quality of life changed during COVID-19*		
Improved	245	12.0
No changed	491	24.1
Deteriorated	1304	63.9
*Diet*		
Healthy diet	102	5.0
Moderately healthy diet	1836	90.0
Unhealthy diet	102	5.0

**Table 2 nutrients-14-00504-t002:** Adherence to a healthy diet according to gender in the pandemic time.

Variables	*n*	%
*Gender—diet*		
Male—healthy diet	18	3.1
Male—moderately healthy diet	515	89.4
Male—unhealthy diet	43	7.4
Female—healthy diet	81	5.5
Female—moderately healthy diet	1328	90.7
Female—unhealthy diet	55	3.7

**Table 3 nutrients-14-00504-t003:** Multinomial logistic regression expressed by risk ratio and standard error.

	Dependent Variable	
	Healthy Diet (1)	Unhealthy Diet (2)
Age Group 31–50	1.176	0.708
	(0.232)	(0.249)
Age Group 51–65	1.302	0.309 *
	(0.393)	(0.611)
Age Group > 65	1.496	0.304
	(0.760)	(1.033)
Gender Male	0.584 *	1.948 ***
	(0.275)	(0.224)
BMI Group 1 Underweight	0.470	1.724
	(0.526)	(0.365)
BMI Group 3 Overweight	0.730	1.320
	(0.278)	(0.257)
BMI Group 4 Obese	0.854	1.380
	(0.365)	(0.374)
Constant	0.063 ***	0.043 ***
	(0.163)	(0.182)
Akaike Inf. Crit.	1572.672	1572.672

Note: * *p* < 0.1; *** *p* < 0.01.

## Data Availability

Not applicable.

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
