# Peer review of "Food Habits and Lifestyle of Romanians in the Context of the COVID-19 Pandemic"

_nutrients, 2022, doi:10.3390/nu14030504_

Round 1
Reviewer 1 Report
First, I would like to thank you for the opportunity to review this article. The subject it deals with is very interesting, since the life habits of the population have been seriously affected by the COVID-19 pandemic and an early detection of risk behaviors is fundamental to be able to design effective intervention policies to prevent critical situations in the long term. However, the manuscript has some shortcomings that require attention and must be corrected for the article to be published. Please consider the suggestions given in this paper to improve the manuscript.
1. Introduction section:
The introduction is too long and there are many aspects that do not coincide with the subject of the study. The authors should restructure it and focus the information on introducing the objective.
2. Materials and Methods:
The authors do not mention:
- The study design.
- There is no section explaining how they recruited the participants.
- Having passed any ethics committee.
3. Results:
- In lines 241- 249 the methodology and results are mixed.
- Figure 1 presents different types of figures that make it difficult to understand.
- The information included in lines 268-271 and 278-281 should be in the discussion section.
4. Discussion:
The authors only summarize the results and do not compare them with other similar studies. They can use the bibliography used for the preparation of the introduction to enrich this section.
In addition, a section referring to the limitations of the study is necessary.
5. Conclusions:
There is not a clear relationship between the results of the study and the conclusion related to “financial impact”.
Author Response
See attached file, thank you.

Reviewer 2 Report
The manuscript entitled ‘Food Habits and Lifestyle of Romanians in the Context of the COVID-19 Pandemic’ presents interesting issue, however some corrections are needed
Abstract:
- Line 18 – a questionnaire with 58 questions’ – the number of items in questionnaire is not necessary (the type or the name of the questionnaire will be preferred)
- Line 29 – A sentence should never start with a number
- Line 30 – A sentence should never start with a number
- Please add specific numeric data accompanied by p-Values in the abstract.
- Please add some conclusions related to the data/ findings (not implication) at the end of the abstract.
Introduction
- Line 61 – ‘has led to the closure of schools in..’ – it was not closure – the remote learning was applied. Please adjust the sentence and the information.
- Lines 70-77 – please add some references to support the information
- The introduction is a little bit too long and some of the information is not strictly related to the main aim of the study. In this section Authors presented the information associated with COVID-19 pandemic mainly in Romania. This section should be briefly presented – what do we know and what is the background for this study. Some detailed information about other studies are necessary. The good background should present the history of problem, the current knowledge and scientific "gap", and then authors should present how their study could fill this gap to justify the study.
Materials and methods
- More detailed information about questionnaire.
- Was the questionnaire previously validated? What were the accuracy and consistency of this questionnaire?
- More information is needed about the validity and reliability of each measure. Additionally, any limitations in reliability and validity need to be addressed in the discussion.
- Please add the information how the questionnaire was distributed? And haw
Results
- ‘Personal and Anthropometric data’ – it should be ‘Sociodemographic and Anthropometric data’
- The share of men in sample is small (28.39%) – this could be a source of bias. It should be discuss and indicated in the limitations of their study.
- Table 1 – one decimal point should be enough
- Table 1 – ‘Sports practitioners during’ – what was a definition of sport activity (time, duration, etc.). similar situation is in case of “diet’ – how healthy diet was defined for respondents?
- Table 3 is a picture (?) why?
- Figure 1 should be improved (readability)
- Figure 4 – please change the fonts
- Figure 5 – please improved the resolution
- I think that some of the data presented in figure should be presented in table (for better readability)
Discussion
- Absolutely too short!!!
- Authors should in their discussion include 3 areas: (1) compare gathered data with the results by other authors, (2) formulate implications of the results of their study and studies by other authors, (3) formulate the future areas which should be studied.
- Authors should present here and discuss the limitations of their study.
Conclusion
- Conclusion section should be improved. Please present the major findings and present the reader what contribution your study has made to the existing literature.
Author Response
See attached file, thank you.

Round 2
Reviewer 2 Report
I appreciate the great efforts that the authors have made in response to my questions and concerns. However, there are some issues that should be corrected:
- (p-valueË‚0.0001) – it should be (pË‚0.0001) – please correct in whole manuscript
- Limitation section – should be presented at the end of the discussion
- There are still some typo (e.g., ‘59. 56.Zaka’ – it should be ‘59. Zaka’)